

# A comprehensive study of the velocity, momentum and position matrix elements for Bloch states: Application to a local orbital basis

**Juan José Esteve-Paredes[1] and Juan José Palacios[1,2]**

**1** Departamento de Física de la Materia Condensada,
Universidad Autónoma de Madrid, E-28049 Madrid, Spain.
**2** Instituto Nicolás Cabrera (INC) and Condensed Matter Physics Center (IFIMAC),
Universidad Autónoma de Madrid, E-28049 Madrid, Spain.

## Abstract

We present a comprehensive study of the velocity operator, $\hat{v} = \frac{i}{\hbar}[\hat{H}, \hat{r}]$, when used in crystalline solids calculations. The velocity operator is key to the evaluation of a number of physical properties and its computation, both from a practical and fundamental perspective, has been a long-standing debate for decades. Our work summarizes the different approaches found in the literature, but never connected before in a comprehensive manner. In particular we show how one can compute the velocity matrix elements following two different routes. One where the commutator is explicitly used and another one where the commutator is avoided by relying on the Berry connection. We work out an expression in the latter scheme to compute velocity matrix elements, generalizing previous results. In addition, we show how this procedure avoids ambiguous mathematical steps and how to properly deal with the two popular gauge choices that coexist in the literature. As an illustration of all this, we present several examples using tight-binding models and local density functional theory calculations, in particular using Gaussian-type localized orbitals as basis sets. We show how the the velocity operator cannot be approximated, in general, by the $k$-gradient of the Bloch Hamiltonian matrix when a non-orthonormal basis set is used. Finally, we also compare with its real-space evaluation through the identification with the canonical momentum operator when possible. This comparison offers us, in addition, a glimpse of the importance of non-local corrections, which may invalidate the naive momentum-velocity correspondence.

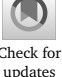
---

# 1  Introduction

The quantum mechanical velocity operator, $\hat{v}$, plays a central role in the evaluation of macroscopic optoelectronic properties of crystalline solids. The velocity matrix elements (VMEs) are generically needed to determine transitions between band states through several formalisms such as Fermi's golden rule [1] for decay and optical excitation processes or the more general Kubo linear response theory [2]. Closely related to $\hat{v}$, the canonical momentum operator $\hat{p}$ also plays a key role, but more from a methodological standpoint. The momentum matrix elements (MME) are, for instance, needed to find parameter-free effective models within $\boldsymbol{k} \cdot \hat{\boldsymbol{p}}$ perturbation theory [3].

It is common to consider the velocity operator as $\hat{v} = i[\hat{H}, \hat{r}]$ (in atomic units, which we will use throughout the text), following a classical to quantum mechanics identification through the Heisenberg equations of motion (Heisenberg picture). In the cases where $\hat{H}$ only contains the kinetic energy and a potential commuting with the position operator $\hat{r}$ (i.e., in absence of spin-orbit coupling or non-local potentials), to work with $\hat{v}$ or with $\hat{p}$ becomes completely equivalent. In coordinate representation this means that $i[H, \boldsymbol{r}]$ and $-i\boldsymbol{\nabla}_{\boldsymbol{r}}$ are interchangeable.

In the following, we review the state of the art of the uses and misuses of these two operators as well as the position operator $\hat{r}$ when evaluating matrix elements between band states. Evaluating the commutator matrix elements presents no problems when dealing with localized states, as in atomic physics, but fundamental difficulties can be found when dealing with Bloch eigenstates due to $\boldsymbol{r}\psi_{n\boldsymbol{k}}(\boldsymbol{r})$ not belonging to the same Hilbert space as that of the states themselves, $\psi_{n\boldsymbol{k}}(\boldsymbol{r})$. This issue has been addressed by Gu and coworkers [4], in addition to presenting an extensive review of the existing $p$-$r$ relations (as called in their work) in the literature. Gu *et al.* find the correct relation between the momentum (or velocity for the case when $\hat{v}$ and $\hat{p}$ are equivalent) and position matrix elements:

$$\langle n\boldsymbol{k}| \hat{\boldsymbol{v}} |n'\boldsymbol{k}'\rangle_v = i[\epsilon_n(\boldsymbol{k}) - \epsilon_{n'}(\boldsymbol{k}')] \langle n\boldsymbol{k}| \hat{\boldsymbol{r}} |n'\boldsymbol{k}'\rangle_v + C_{n\boldsymbol{k}, n'\boldsymbol{k}'}. \tag{1}$$

When deriving the previous formula, eigenstates are normalized following the usual mathematical convention of considering a finite volume $v$ and making wavefunctions obey periodic boundary conditions (PBCs), therefore not decaying at the boundaries even if the volume tends

to infinity. The surface term $C$ is calculated on the (hypothetical) surface of the solid. With this is mind, one can easily convince oneself that the position (also called dipole) matrix element depends on the origin of coordinates and that $C$ compensates this choice (as the VME cannot depend on the origin). Eq. (1) above presents a fundamental view of $i[\hat{H},\hat{r}]$ rather than convenient shortcut to evaluate the VME, as a challenging integration in coordinate space is needed on the right side of the equality. It also remarks the difference with the atomic case, where the surface term does not appear.

In practice, Bloch eigenstates are very often represented in a Bloch basis which, in turn, may be expanded in a local orbital basis. In this regard, a good effort has been put in the actual evaluation of the VME over the last decades. As we will show below, the VME between same-$k$ Bloch eigenstates can be calculated through the following expression:

$$
\begin{aligned}
\langle n\boldsymbol{k}|\hat{v}|n'\boldsymbol{k}\rangle_{v} = & \sum_{\alpha\alpha'} c_{\alpha n}^{*}(\boldsymbol{k})c_{\alpha'n'}(\boldsymbol{k})\nabla_{\boldsymbol{k}}\langle\alpha\boldsymbol{k}|\hat{H}|\alpha'\boldsymbol{k}\rangle_{v} \\
& + \sum_{\alpha\alpha'} c_{\alpha n}^{*}(\boldsymbol{k})c_{\alpha'n'}(\boldsymbol{k})\Big[i\epsilon_{n}(\boldsymbol{k})A_{\alpha\alpha'}(\boldsymbol{k}) - i\epsilon_{n'}(\boldsymbol{k})A_{\alpha'\alpha}^{*}(\boldsymbol{k})\Big],
\end{aligned}
\tag{2}
$$

where the $c$'s are the coefficients of the expansion of eigenstates in a generic $|\alpha\boldsymbol{k}\rangle_v$ Bloch basis and $A_{\alpha\alpha'}(\boldsymbol{k})$ is the Berry connection associated with such a Bloch basis. There is a vast literature where one can find expressions seemingly related to Eq. (2) with implementations involving local orbital basis sets. Our derivation of Eq. (2) does not need the aid of these type of basis sets, thus constituting a generalization of previous works.

The first term of Eq. (2) is sometimes referred to as the Peierls approximation (we will also call it the "$k$-gradient approximation"), while the second term is needed to deliver the full matrix element. The suppression of the second and third terms, as we will show later, can lead to large and uncontrolled quantitative errors. This issue was first explored by Pedersen *et al.* [5], by trying to complete the Peierls approximation within a tight-binding scheme. Paul *et al.* [6] noted the importance of using the Peierls approximation with the appropiate gauge choice in a Wannier basis. Later, Wang *et al.* [7] derived the equivalent of Eq. (2) for the specific case of a Wannier basis with the aid of perturbation theory. Years after, Tomczak *et al.* [8] independently worked out an expression for VMEs, introducing an intra-unit cell correction to $\nabla_{k}H_{\alpha\alpha'}(\boldsymbol{k})$ (we will see later that this can be understood as a consequence of the gauge choice), and adding extra terms to this quantity similar to those in Eq. (2). Tomczak *et al.* contribution was later replicated, perhaps in a clearer and more complete way, in the work of Lee et al. [9], who presented a complete expression similar to that of Eq. (2) for a general, nonorthonormal local basis. Actually, as originally reported in Ref. [9], the Berry connection did not appear. The fact that Eq. (2) can be recast in this form will be shown below in this work, thus generalizing the evaluation of the VME to any basis, not necessarily comprised of local orbitals. The relative importance of every term in Eq. (2) is an open issue, mainly because it depends on the specific basis and gauge choice that is used. We note a very recent work by Ibanez-Aspiroz *et al.* [10] exploring this issue in the context of a Wannier-function interpolation scheme. They have found that neglecting inter-atomic contributions when evaluating VMEs can lead to important errors in the evaluation of optical properties. Here we perform a similar analysis by means of density functional theory (DFT) calculations relying on Gaussian-type orbitals (GTOs).

In the light of Eq. (1), Eq. (2) presents a somewhat puzzling aspect: first, there is no surface term and, second, no term depends on the placement of the integration volume $v$. However, Eq. (2) was derived in Ref. [9] essentially in the same manner as Eq. (1) was derived in Ref. [4], namely, by making use of $\hat{v} = i[\hat{H},\hat{r}]$ projected in a chosen basis, the coordinate basis in Eq. (1) and a Bloch basis (in turn, constructed from a local orbital basis) in Eq. (2). This have has been unnoticed in the literature and is an additional motivation to

carry out the present work. We will also explain how the the popular expression for position matrix elements given by Blount [11] fits into this comparison.

We organize this study as follows. In Sec. 2 we present the main theoretical ingredients by first recalling the differences between periodic boundary conditions versus the infinite volume case when defining Bloch eigenstates. We follow by introducing two ways of treating $\hat{v} = i[\hat{H}, \hat{r}]$, one relying on an integration in the whole finite volume of the system, and the other based on using the $k$ representation for operators, involving matrix elements between the cell-periodic part of the Bloch eigenstates. In Sec. 3 we explain how Eq. (2) rigorously comes about from the second method, while showing the way it has been previously derived in the literature is mathematically inconsistent, to say the least. In Sec. 4 we present a numerical study that gives us insight into the quantitative error that one makes when assuming the equality $\langle n\mathbf{k}|\hat{v}|n'\mathbf{k}'\rangle = \langle n\mathbf{k}|\hat{p}|n'\mathbf{k}'\rangle$ (again, recall the use of atomic units), even in the presence of a local potential, and the trade-off between computational simplicity and accuracy when using the Peierls approximation in a practical situation. Finally, we summarize our main conclusions in Sec. 5.

## 2 Theory

### 2.1 Preliminary definitions

We start by recalling the normalization choices for eigenstates in a crystal. This turns out to be a key point to understand the relation between position and velocity operators. Two options are compatible with Bloch theorem: one can assume a finite volume $v$ normalization or let the eigenstates extend to all space following a distribution-like normalization. In order to distinguish the two cases, we write

$$|n\mathbf{k}\rangle_v = \frac{1}{\sqrt{N}} e^{i\mathbf{k}\cdot\hat{r}} |u_{n\mathbf{k}}\rangle \,, \quad |n\mathbf{k}\rangle = \frac{1}{(2\pi)^{2/3}} e^{i\mathbf{k}\cdot\hat{r}} |u_{n\mathbf{k}}\rangle \,, \tag{3}$$

being the normalization conditions for every case case

$$
\begin{aligned}
\langle n\mathbf{k}|n'\mathbf{k}'\rangle_v &= \int_{v[\mathbf{x}_0]} d^3r\, \psi_{n\mathbf{k}}^{(v)*}(\mathbf{r})\psi_{n'\mathbf{k}'}^{(v)}(\mathbf{r}) = \delta_{nn'}\delta_{\mathbf{k}\mathbf{k}'} \,, \\
\langle n\mathbf{k}|n'\mathbf{k}'\rangle &= \int_{-\infty}^{\infty} d^3r\, \psi_{n\mathbf{k}}^{*}(\mathbf{r})\psi_{n'\mathbf{k}'}(\mathbf{r}) = \delta_{nn'}\delta(\mathbf{k}-\mathbf{k}') \,.
\end{aligned}
\tag{4}
$$

Note that this convention assumes $|u_{n\mathbf{k}}\rangle$ to be normalized to one in the finite volume case and to the unit cell volume (denoted with $\Omega$ in the following) in the distribution case. The $\int_{v[\mathbf{x}_0]}$ means that the integration domain (the volume $v$) is defined by the parallelepiped determined by $N_i \mathbf{a}_i$ with $N_i$ being the number of cells in each direction given by the primitive vectors $\mathbf{a}_i$. Its origin vertex is located at the $\mathbf{x}_0$ point, which have to be selected in the first place. Finally $N = N_1 N_2 N_3$ is the total number of cells of the crystal. Born-von Karman boundary conditions $\psi_{n\mathbf{k}}(\mathbf{r} + N_i \mathbf{a}_i) = \psi_{n\mathbf{k}}(\mathbf{r})$ are applied, leading to a quantization of the crystal momentum according to $\mathbf{k} = \frac{l_1}{N_1}\mathbf{G}_1 + \frac{l_2}{N_2}\mathbf{G}_2 + \frac{l_3}{N_3}\mathbf{G}_3$ with $\mathbf{G}$ being reciprocal lattice vectors and $l_i = -M_i, \ldots, M_i$ such as $N_i = 2M_i + 1$. In general, matrix elements of operators whose application on eigenstates breaks periodicity may depend on $\mathbf{x}_0$. On the other hand, in the case of infinite volume normalization, the integrals run over the whole unbounded space, including infinities. In this case, $\mathbf{k}$ vectors become a dense set inside the Brillouin zone (BZ).

The representation of a given operator $\hat{O}$ in both cases becomes (we will assume that $\int$ is

equivalent to $\int_{-\infty}^{\infty}$ in the following)

$$O_{n\mathbf{k},n\mathbf{k}'}^{(v)} \equiv \langle n\mathbf{k}|\hat{O}|n'\mathbf{k}'\rangle_v = \int_{v[\mathbf{x}_0]} d^3r\, \psi_{n\mathbf{k}}^{(v)*}(\mathbf{r})[\hat{O}\psi_{n'\mathbf{k}'}^{(v)}](\mathbf{r}),$$

$$O_{n\mathbf{k},n'\mathbf{k}'} \equiv \langle n\mathbf{k}|\hat{O}|n'\mathbf{k}'\rangle = \int d^3r\, \psi_{n\mathbf{k}}^*(\mathbf{r})[\hat{O}\psi_{n'\mathbf{k}'}](\mathbf{r}),$$

(5)

where $[\hat{O}\psi](\mathbf{r}) \equiv \langle \mathbf{r}|\hat{O}|\psi\rangle$. If the operator $\hat{O}$ is such that $[\hat{O}\psi](\mathbf{r})$ is still of Bloch form, the matrix elements can be reduced to an integration within the unit cell involving the periodic part of eigenstates

$$\langle n\mathbf{k}|\hat{O}|n'\mathbf{k}'\rangle_v = \delta_{\mathbf{k}\mathbf{k}'}\, \langle u_{n\mathbf{k}}|\hat{O}_{\mathbf{k}}|u_{n'\mathbf{k}}\rangle_\Omega\,,$$

(6)

where $\hat{O}_{\mathbf{k}} \equiv e^{-i\mathbf{k}\cdot\hat{\mathbf{r}}}\hat{O}e^{i\mathbf{k}\cdot\hat{\mathbf{r}}}$, sometimes called the "$k$ representation of an operator". The Kronecker delta is factorize using the relation $\frac{1}{N}\sum_{\mathbf{R}} e^{i(\mathbf{k}-\mathbf{k}')\cdot\mathbf{R}} = \delta_{\mathbf{k}\mathbf{k}'}$ for wave vectors inside the first BZ, after doing the usual change from the total integration volume to a sum of unit cell volumes. In the infinite volume case the expression is similar but taking $u_{n\mathbf{k}}(\mathbf{r}) \longrightarrow \Omega^{-1/2}u_{n\mathbf{k}}(\mathbf{r})$, according to our criteria of Eq. (3), as well as replacing the Kronecker delta by a Dirac delta. In what follows we particularize to the velocity operator $\hat{\mathbf{v}}$ and its relation to other quantities.

## 2.2 Relation between the velocity and momentum matrix elements

As discussed in the introduction, the velocity and momentum operators can only be interchanged if spin-orbit coupling is neglected and the periodic potential in the crystal is assumed to be local. Let the Hamiltonian be separated into $\hat{H} = \hat{H}_L + \hat{H}'$, where $\hat{H}_L = \hat{\mathbf{p}}^2/2 + V(\hat{\mathbf{r}})$ with $V(\hat{\mathbf{r}})$ the local periodic part of the lattice potential. $\hat{H}'$ includes all the non-local parts of the Hamiltonian, e.g. pseudopotential terms, spin-orbit coupling, or even the contribution of non-local functionals in case they are used in the DFT calculation. Then, for the first term one can write $\hat{\mathbf{p}} = i[\hat{H}_L, \hat{\mathbf{r}}]$, and the projection of the full velocity operator on a subspace of band states can be written as

$$\langle n\mathbf{k}|\hat{\mathbf{v}}|n'\mathbf{k}'\rangle_v = i\,\langle n\mathbf{k}|[\hat{H}_L + \hat{H}', \hat{\mathbf{r}}]|n'\mathbf{k}'\rangle_v = \langle n\mathbf{k}|\hat{\mathbf{p}}|n'\mathbf{k}'\rangle_v + i\,\langle n\mathbf{k}|[\hat{H}', \hat{\mathbf{r}}]|n'\mathbf{k}'\rangle_v\,.$$

(7)

The presence of the second term in Eq. (7) is challenging from a practical standpoint and only when $\hat{H}' = 0$, Eq. (7) becomes the theoretical velocity-momentum equality. In any case, evaluating the VME seems to require, in principle, the evaluation of the MME through its representation $-i\nabla_{\mathbf{r}}$.

In a practical calculation, one can expect an appreciable discrepancy between the left and right hand sides of the equation above. This is due to the closure relation $\hat{I} = \sum_{n\mathbf{k}} |n\mathbf{k}\rangle\langle n\mathbf{k}|$ not being fully satisfied, as the Hilbert space is truncated in first principles calculations. This can also be ultimately traced back to the fact that the canonical commutation relation $[\hat{r}_\alpha, \hat{p}_\beta] = i\delta_{\alpha\beta}$ can never exactly hold for a finite-matrix representation since in such cases $\text{Tr}(\hat{r}_\alpha \hat{p}_\beta) = \text{Tr}(\hat{p}_\alpha \hat{r}_\beta)$. Therefore, one can only expect the (7) to be negligible if the physical states are sufficiently well represented in the working Hilbert space. We will give below a few examples of this practical limitation.

In the following we explore two routes that can be followed to by-pass the evaluation of the MME and, at the same time, of the non-local term if present.

## 2.3 Relation between velocity matrix elements and the Berry connection

We first write the VME in the $k$ representation. It is easy to see that $\hat{\mathbf{v}}_{\mathbf{k}} = e^{-i\mathbf{k}\cdot\hat{\mathbf{r}}}i[\hat{H}, \hat{\mathbf{r}}]e^{i\mathbf{k}\cdot\hat{\mathbf{r}}} = \nabla_{\mathbf{k}}\hat{H}_{\mathbf{k}}$, so one can write

$$\langle n\mathbf{k}|\hat{\mathbf{v}}|n'\mathbf{k}'\rangle_v = \delta_{\mathbf{k}\mathbf{k}'}\, \langle u_{n\mathbf{k}}|(\nabla_{\mathbf{k}}\hat{H}_{\mathbf{k}})|u_{n'\mathbf{k}}\rangle_\Omega\,.$$

(8)

By applying chain rule it is straightforward to find

$$\langle n\boldsymbol{k}|\hat{\boldsymbol{v}}|n'\boldsymbol{k}'\rangle_v = \delta_{\boldsymbol{k}\boldsymbol{k}'}[i\omega_{n\boldsymbol{k},n'\boldsymbol{k}}\boldsymbol{A}_{nn'}(\boldsymbol{k}) + \nabla_{\boldsymbol{k}}\epsilon_n(\boldsymbol{k})\delta_{nn'}], \tag{9}$$

where $\omega_{n\boldsymbol{k},n'\boldsymbol{k}} \equiv \epsilon_n(\boldsymbol{k}) - \epsilon_{n'}(\boldsymbol{k})$ and $\boldsymbol{A}_{nn'}(\boldsymbol{k}) \equiv i\langle u_{n\boldsymbol{k}}|\nabla_{\boldsymbol{k}}u_{n'\boldsymbol{k}}\rangle_{\Omega}$, this last quantity being the Berry connection.[1] Eq. (9) can be found in the literature, see e.g. Ref. [12]. The equation above replaces Eq. (7) by introducing the evaluation of the Berry connection associated with the Bloch eigenstates. This, however, can be a cumbersome task since $k$ derivatives of eigenstates are not known in numerical diagonalization procedures. While this problem can be circumvented through perturbation theory [13], in Sec. 3 we show how Eq. (9) can be recast in a more convenient and familiar form. Incidentally, note that Eq. (9) provides a way to compute the non-diagonal Berry connection elements if the VMEs are known.

## 2.4 Relation between velocity and position matrix elements

Alternatively, we can directly perform the integrals that appear in Eq. (7) when representing in coordinate space. Assuming $\hat{H}' = 0$ and, therefore, being able to write $\hat{\boldsymbol{v}} \equiv i[\hat{H},\hat{\boldsymbol{r}}] = \hat{\boldsymbol{p}}$, one is free to use $-i\nabla_{\hat{\boldsymbol{r}}}$ or $i[\hat{H},\hat{\boldsymbol{r}}]$. In both cases the explicit knowledge of the real-space wavefunction of the eigenstates is required. In the former case derivatives need to be carried out, which depending on the orbital basis can be more or less cumbersome to implement. In the latter, the use of the commutator entails further steps, where one needs to pay attention to the correct use of the hermiticity of $\hat{H}$ in the $\hat{H}\hat{\boldsymbol{r}}$ product. This procedure, which has been followed by Gu and coworkers in Ref. [4], only applies to eigenstates in the framework of finite volume normalization, where integrals for matrix elements can be converged. One starts with

$$\langle n\boldsymbol{k}|\hat{\boldsymbol{v}}|n'\boldsymbol{k}'\rangle_v = i\int_{v[\boldsymbol{x}_0]}d^3r\,\psi^{(v)*}_{n\boldsymbol{k}}(\boldsymbol{r})[H\boldsymbol{r} - \boldsymbol{r}H]\psi^{(v)}_{n'\boldsymbol{k}'}(\boldsymbol{r}), \tag{10}$$

where we have to act with $-\nabla^2_{\boldsymbol{r}}$ on $\boldsymbol{r}\psi^{(v)}_{n'\boldsymbol{k}'}(\boldsymbol{r})$ and $\psi^{(v)}_{n'\boldsymbol{k}'}(\boldsymbol{r})$. After performing the derivatives and using Gauss's theorem, one obtains

$$\langle n\boldsymbol{k}|\hat{\boldsymbol{v}}|n'\boldsymbol{k}'\rangle_v = i\omega_{n\boldsymbol{k},n'\boldsymbol{k}'}\langle n\boldsymbol{k}|\hat{\boldsymbol{r}}|n'\boldsymbol{k}'\rangle_v + \boldsymbol{C}_{n\boldsymbol{k},n'\boldsymbol{k}'}, \tag{11}$$

where

$$\boldsymbol{C}_{n\boldsymbol{k},n'\boldsymbol{k}'} = -\frac{i}{2}\int_{\partial v[\boldsymbol{x}_0]}d\boldsymbol{S}\cdot\left\{\psi^{(v)*}_{n\boldsymbol{k}}(\boldsymbol{r})\nabla_{\boldsymbol{r}}\psi^{(v)}_{n'\boldsymbol{k}'}(\boldsymbol{r}) - [\nabla_{\boldsymbol{r}}\psi^{(v)}_{n\boldsymbol{k}}(\boldsymbol{r})]^*\psi^{(v)}_{n'\boldsymbol{k}'}(\boldsymbol{r})\right\}\boldsymbol{r} \tag{12}$$

(same comment[1] applies here). Note the presence of the $\boldsymbol{r} = (x,y,z)$ breaking the periodicity of the integrand at two opposite surfaces. The appearance of this last surface term arises from the finite value of the wavefunctions in the surface of the material volume, which we denote with $\partial v[\boldsymbol{x}_0]$. It is important to note that the wavefunctions do not decay even in the limit of an infinite volume and this term is always present.

As noticed in Ref. [4], the hermiticity property cannot be applied as usual in $\langle n\boldsymbol{k}|\hat{\boldsymbol{v}}|n'\boldsymbol{k}'\rangle_v = i\langle n\boldsymbol{k}|[\hat{H}\hat{\boldsymbol{r}} - \hat{\boldsymbol{r}}\hat{H}]|n'\boldsymbol{k}'\rangle_v$, which results in the surface term above. Secondly, both matrix elements on the right hand side (RHS) in Eq. (11) depend on the origin of the integration volume and are not $k$-diagonal, while the sum does not depend on this arbitrary choice of origin and is diagonal in the wave vector as the VME actually is. The relative weight of $\langle n\boldsymbol{k}|\hat{\boldsymbol{r}}|n'\boldsymbol{k}'\rangle_v$ versus $\boldsymbol{C}_{n\boldsymbol{k},n'\boldsymbol{k}'}$ with respect to the full VME is also explored in Ref. [4] showing that, in general, one cannot find a point $\boldsymbol{x}_0$ that makes the surface term to vanish, even for certain analytical models.

---

[1] Note that Berry connection is not an operator and hence it does not follow the initial definition of Eq. (5).

We note that in the diagonal case, $\langle n\boldsymbol{k}|\hat{\boldsymbol{v}}|n\boldsymbol{k}\rangle$, the first term of Eq. (10) vanishes, and all contribution goes to $\boldsymbol{C}_{nk,nk}$. In that case the dependence of this term with the origin for the crystal is removed due to the symmetry inside the integrand. Comparing with Eq. (9), we see that the value of $\boldsymbol{C}_{nk,nk}$ is equal to $\nabla_{\boldsymbol{k}}\epsilon_n(\boldsymbol{k})$.

Therefore, Eq. (11) presents no advantage versus directly computing $-i\langle n\boldsymbol{k}|\nabla_{\hat{r}}|n'\boldsymbol{k}'\rangle$ to find the VME, as one still has to perform nontrivial integrations for position and surface matrix elements. It provides us, however, with the conclusion that momentum and dipole matrix elements (multiplied by the frequency) should never be interchanged when dealing with Bloch eigenstates in a finite volume.

## 2.5 Relation between velocity and position matrix elements with a distribution basis

If Bloch eigenstates are normalized as distributions [recall Eqs. (3) and (4)], then one can still use them as a basis to represent general physical quantum states in the crystal. We will name this basis states as the distribution basis. This was originally referred to as the crystal momentum representation (CMR) [11], where one writes

$$|\phi\rangle = \sum_n \int_{\text{BZ}} d^3k\, g_n(\boldsymbol{k})|n\boldsymbol{k}\rangle\,, \tag{13}$$

with $g_n(\boldsymbol{k})$ being the envelope function for the $n$ band. The matrix elements between two physical states is written

$$\langle \phi_1|\hat{O}|\phi_2\rangle = \sum_{nn'} \int_{\text{BZ}} d^3k\, d^3k'\, g_n^{(1)*}(\boldsymbol{k}) g_{n'}^{(2)}(\boldsymbol{k}') \langle n\boldsymbol{k}|\hat{O}|n'\boldsymbol{k}'\rangle\,. \tag{14}$$

Now one needs to find the matrix element $\langle n\boldsymbol{k}|\hat{O}|n'\boldsymbol{k}'\rangle$ that enters in the calculation above. As only the full $n$ sums and $k$ integrations are relevant, we can take into account the boundary properties of the state $|\phi\rangle$. This is the case of the matrix elements for the position operator, for which Blount [11] noticed that $\langle n\boldsymbol{k}|\hat{\boldsymbol{r}}|n'\boldsymbol{k}'\rangle$ is ill-defined by itself but that a distribution form can be given if $\hat{\boldsymbol{r}}$ is assumed to act on a state $|\phi\rangle$ belonging to its domain. Specifically, Blount showed that

$$\langle n\boldsymbol{k}|\hat{\boldsymbol{r}}|n'\boldsymbol{k}'\rangle = -i\nabla_{\boldsymbol{k}'}\delta(\boldsymbol{k}'-\boldsymbol{k})\delta_{nn'} + \frac{1}{\Omega}\delta(\boldsymbol{k}'-\boldsymbol{k})\boldsymbol{A}_{nn'}(\boldsymbol{k})\,. \tag{15}$$

The effect of boundary conditions on eigenstates is highlighted here, as $\langle n\boldsymbol{k}|\hat{\boldsymbol{r}}|n'\boldsymbol{k}'\rangle$ fundamentally differs from $\langle n\boldsymbol{k}|\hat{\boldsymbol{r}}|n'\boldsymbol{k}'\rangle_v$, addressed in the previous section. Here, only the diagonal matrix elements of the position operator depend on an origin through the arbitrary choice $\hat{\boldsymbol{r}} \to \hat{\boldsymbol{r}} + \boldsymbol{d}$, which involves doing $\langle n\boldsymbol{k}|\hat{\boldsymbol{r}}|n'\boldsymbol{k}'\rangle \to \langle n\boldsymbol{k}|\hat{\boldsymbol{r}}|n'\boldsymbol{k}'\rangle + \boldsymbol{d}\,\delta_{nn'}\delta(\boldsymbol{k}-\boldsymbol{k}')$. On the other hand, assuming a finite volume always makes $\langle n\boldsymbol{k}|\hat{\boldsymbol{r}}|n'\boldsymbol{k}'\rangle_v$ depend on the integration limits. Position operator matrix elements in Eq. (15) do not depend on any arbitrary origin, but only make sense within Eq. (14).

As far as the velocity operator is concerned, Eq. (9) is still perfectly valid in the infinite volume case:

$$\langle n\boldsymbol{k}|\hat{\boldsymbol{v}}|n'\boldsymbol{k}'\rangle = \delta(\boldsymbol{k}-\boldsymbol{k}')[\nabla_{\boldsymbol{k}}\epsilon_n(\boldsymbol{k})\delta_{nn'} + \frac{i}{\Omega}\omega_{nk,n'k'}\boldsymbol{A}_{nn'}(\boldsymbol{k})]\,, \tag{16}$$

expression to be used, again, only in the context of Eq. (14). Alternatively, in Appendix A we also show that projecting $\hat{\boldsymbol{v}}$ on general physical states $\langle \phi|\hat{\boldsymbol{v}}|\phi'\rangle = i\langle \phi|[\hat{H},\hat{\boldsymbol{r}}]|\phi'\rangle$, along with Eq. (15), also leads to Eq. (16).

Finally, to complete the connection between the different matrix element expressions, it is straightforward to show that

$$\langle n\mathbf{k}|\hat{\mathbf{v}}|n'\mathbf{k}'\rangle = i\omega_{n\mathbf{k},n'\mathbf{k}'}\langle n\mathbf{k}|\hat{\mathbf{r}}|n'\mathbf{k}'\rangle. \tag{17}$$

Again, one simply needs to project $\langle\phi|\hat{\mathbf{v}}|\phi'\rangle = i\langle\phi|[\hat{H},\hat{\mathbf{r}}]|\phi'\rangle$ and proceed in the same manner as explained in previous section. Here, however, the surface term vanishes due to $|\phi\rangle$ and $|\phi'\rangle$ being square-integrable over all space. Notice that this momentum and position relation matches that in atomic physics. This is also shown in Ref. [4] by using narrow $k$ envelope functions in the limit of zero width.

# 3 Velocity matrix elements when representing in a Bloch basis.

Having established a comprehensive overview of the available recipes to evaluate the VME and their proper use, we proceed now with their actual computation when a generic and possibly non-orthonormal Bloch basis is used to expand the Bloch eigenstates:

$$|n\mathbf{k}\rangle_v = \sum_\alpha c_{\alpha n}(\mathbf{k})|\alpha\mathbf{k}\rangle_v. \tag{18}$$

We stress again that $|\alpha\mathbf{k}\rangle_v$ is a generic basis state satisfying Bloch's theorem in a finite volume, with $\alpha$ being a generic quantum number. The coefficients $c_{\alpha n}(\mathbf{k})$ are found by solving the generalized eigenvalue problem

$$\sum_{\alpha'} H_{\alpha\alpha'}^{(v)}(\mathbf{k})c_{\alpha'n}(\mathbf{k}) = \epsilon_n(\mathbf{k})\sum_{\alpha'} S_{\alpha\alpha'}^{(v)}(\mathbf{k})c_{\alpha'n}(\mathbf{k}), \tag{19}$$

where $H_{\alpha\alpha'}^{(v)}(\mathbf{k})$ and $S_{\alpha\alpha'}^{(v)}(\mathbf{k})$ are the matrices representing the Hamiltonian and identity operators, respectively.

Eq. (9) can now be properly converted into more familiar expression. First the Berry connection reads

$$A_{nn'}(\mathbf{k}) = i\sum_{\alpha\alpha'} S_{\alpha\alpha'}^{(v)}(\mathbf{k})c_{\alpha n}^*(\mathbf{k})\nabla_\mathbf{k}c_{\alpha'n'}(\mathbf{k}) + \sum_{\alpha\alpha'} c_{\alpha n}^*(\mathbf{k})c_{\alpha'n'}A_{\alpha\alpha'}(\mathbf{k}). \tag{20}$$

In this expression one has to perform derivatives in $k$ space of the coefficients $c_{\alpha n}(\mathbf{k})$. In most cases these coefficients are obtained by numerical diagonalization of Eq. (19) so that they are not continuous and, therefore, differentiable. However, this can be avoided by directly employing the chain rule after inserting Eq. (20) into Eq. (9), leading to

$$\begin{aligned}
\langle n\mathbf{k}|\hat{\mathbf{v}}|n'\mathbf{k}\rangle_v &= \mathbf{v}_{nn'}^{(A)}(\mathbf{k}) + \mathbf{v}_{nn'}^{(B)}(\mathbf{k}); \\
\mathbf{v}_{nn'}^{(A)}(\mathbf{k}) &= \sum_{\alpha\alpha'} c_{\alpha n}^*(\mathbf{k})c_{\alpha'n'}(\mathbf{k})\nabla_\mathbf{k}H_{\alpha\alpha'}^{(v)}(\mathbf{k}), \\
\mathbf{v}_{nn'}^{(B)}(\mathbf{k}) &= i\sum_{\alpha\alpha'} c_{\alpha n}^*(\mathbf{k})c_{\alpha'n'}(\mathbf{k})\left[\epsilon_n(\mathbf{k})A_{\alpha\alpha'}(\mathbf{k}) - \epsilon_{n'}(\mathbf{k})A_{\alpha'\alpha}^*(\mathbf{k})\right].
\end{aligned} \tag{21}$$

We show the complete derivation in Appendix B. Eq. (21) is one important result of this work: it allows to compute VME from the Hamiltonian matrix elements and the Berry connection in whichever Bloch basis. It also generalizes similar formulas that can be found in the literature [see our discussion at the introduction section about Eq.(2)]. We have differentiated two contributions, A and B, to the VME. The first one is evokes the exact expression $\hat{\mathbf{v}}_\mathbf{k} = \nabla_\mathbf{k}\hat{H}_\mathbf{k}$, but the second one is equally important, as we will show below. Eq. (21) clearly shows that $\nabla_\mathbf{k}\hat{H}_\mathbf{k}$ is not, in general, equivalent to $\nabla_\mathbf{k}H_{\alpha\alpha'}^{(v)}(\mathbf{k})$.

In many practical cases the Bloch basis is expanded, in turn, in a local orbital basis. Regarding this, two different types of basis can be found in the literature:

$$|\alpha \boldsymbol{k}\rangle_v = \frac{1}{\sqrt{N}} \sum_{\boldsymbol{R}} e^{i\boldsymbol{k}\cdot\boldsymbol{R}} |\alpha \boldsymbol{R}\rangle_v \quad \text{and} \quad |\tilde{\alpha} \boldsymbol{k}\rangle_v = \frac{1}{\sqrt{N}} \sum_{\boldsymbol{R}} e^{i\boldsymbol{k}\cdot(\boldsymbol{R}+\boldsymbol{d}_\alpha)} |\alpha \boldsymbol{R}\rangle_v, \tag{22}$$

where $|\alpha \boldsymbol{R}\rangle_v$ is an orbital with $\boldsymbol{d}_\alpha$ position vector inside the unit cell of site $\boldsymbol{R}$. A finite size crystal containing $N$ cells is assumed throughout. Bloch eigenstates are now given by

$$|n\boldsymbol{k}\rangle_v = \sum_\alpha c_{\alpha n}(\boldsymbol{k})|\alpha \boldsymbol{k}\rangle_v \quad \text{or likewise} \quad |n\boldsymbol{k}\rangle_v = \sum_\alpha b_{\alpha n}(\boldsymbol{k})|\tilde{\alpha} \boldsymbol{k}\rangle_v, \tag{23}$$

with both expansions being related by $c_{\alpha n}(\boldsymbol{k}) = e^{i\boldsymbol{k}\cdot\boldsymbol{d}_\alpha} b_{\alpha n}(\boldsymbol{k})$. We will use the former basis in this work by default, which we refer to as the *cell gauge*, and make considerations related to the other one, the *atom gauge*, when appropriate. In this basis the matrices needed in Eq. (19) become

$$H_{\alpha\alpha'}^{(v)}(\boldsymbol{k}) = \sum_{\boldsymbol{R}} e^{i\boldsymbol{k}\cdot\boldsymbol{R}} \langle \alpha\boldsymbol{0}|\hat{H}|\alpha'\boldsymbol{R}\rangle \quad \text{and} \quad S_{\alpha\alpha'}^{(v)}(\boldsymbol{k}) = \sum_{\boldsymbol{R}} e^{i\boldsymbol{k}\cdot\boldsymbol{R}} \langle \alpha\boldsymbol{0}|\alpha'\boldsymbol{R}\rangle. \tag{24}$$

Now we can recast the Berry connection terms in Eq. (21) into an explicit form involving the position operator

$$\begin{aligned}
\boldsymbol{A}_{\alpha\alpha'}(\boldsymbol{k}) &= \sum_{\boldsymbol{R}} e^{i\boldsymbol{k}\cdot\boldsymbol{R}} \langle \alpha\boldsymbol{0}|\hat{\boldsymbol{r}}|\alpha'\boldsymbol{R}\rangle + i\nabla_{\boldsymbol{k}} S_{\alpha\alpha'}^{(v)}(\boldsymbol{k}), \\
\boldsymbol{A}_{\alpha'\alpha}^*(\boldsymbol{k}) &= \sum_{\boldsymbol{R}} e^{i\boldsymbol{k}\cdot\boldsymbol{R}} \langle \alpha\boldsymbol{0}|\hat{\boldsymbol{r}}|\alpha'\boldsymbol{R}\rangle.
\end{aligned} \tag{25}$$

Note that Berry connection between non-orthonormal states is not an hermitian quantity and present a certain asymmetry for its conjugate. The gradient of the overlap matrix can be directly computed using Eq. (19). With this expressions, Eq. (21) becomes identical to that reported in Ref. [9] (we invite the reader to see Appendix B for all the details in the derivation).

It is important to note that neither of the two terms in Eq. (21) is gauge independent. One can easily check how the two terms change when switching to the atom gauge, according to Eq. (22). For instance, the first term becomes (we denote the gauge choice in the superscripts)

$$\begin{aligned}
\boldsymbol{v}_{nn'}^{(A,\,\text{atom})}(\boldsymbol{k}) &= \sum_{\alpha\alpha'} b_{\alpha n}^*(\boldsymbol{k}) b_{\alpha'n'}(\boldsymbol{k}) \nabla_{\boldsymbol{k}} \tilde{H}_{\alpha\alpha'}^{(v)}(\boldsymbol{k}) \\
&= \boldsymbol{v}_{nn'}^{(A,\,\text{cell})}(\boldsymbol{k}) + i \sum_{\alpha\alpha'} c_{\alpha n}^*(\boldsymbol{k}) c_{\alpha'n'}(\boldsymbol{k}) H_{\alpha\alpha'}^{(v)}(\boldsymbol{k})(\boldsymbol{d}_{\alpha'} - \boldsymbol{d}_\alpha),
\end{aligned} \tag{26}$$

while the correction for the B term is the same with opposite sign, showing that the absolute value of the sum $\boldsymbol{v}_{nn'}^{(A)} + \boldsymbol{v}_{nn'}^{(B)}$ is gauge invariant, as it should be for a physical operator.

It is an interesting exercise to obtain the form of Eq. (21) when one works with maximally localized Wannier functions (MLWFs) and in the purely tight-binding (TB) limit. MLWFS still have a finite spread while TB orbitals are considered to be point-like. Despite this difference, one usually neglects inter-atomic and intra-atomic position matrix elements beyond orbital centers [10] in both cases. Denoting this basis orbitals with $v$, we obtain

$$\langle n\boldsymbol{k}|\hat{\boldsymbol{v}}|n'\boldsymbol{k}\rangle_v = \sum_{vv'} c_{vn}^*(\boldsymbol{k}) c_{v'n'}^*(\boldsymbol{k}) \nabla_{\boldsymbol{k}} H_{vv'}^{(v)}(\boldsymbol{k}) + i \sum_{vv'} c_{vn}^*(\boldsymbol{k}) c_{v'n'}^*(\boldsymbol{k}) H_{vv'}^{(v)}(\boldsymbol{k})(\boldsymbol{d}_{v'} - \boldsymbol{d}_v), \tag{27}$$

which is the same as Eq. (26). This tells us that if we neglect inter-orbital contributions, the computation of the VME with only the gradient term (the A term) in the atom gauge, therefore neglecting $\boldsymbol{v}_{nn'}^{(B,\,\text{atom})}$, is equivalent to computing both terms (the full VME) in the cell gauge.

This means that $\boldsymbol{v}_{nn'}^{(B,\,\text{atom})} = 0$, as can be easily checked. The second line of Eq. (26) [or equivalently Eq. (27)] was presented in Ref. [8] as a "Peierls substitution approach to the case of multiatomic unit cells". Based on our previous discussion, we see that it appears naturally within the atom gauge. In the more general case of a non-orthonormal basis, both terms of Eq. (21) must be evaluated regardless of the gauge choice. We examine this more in depth in Sec. 4 by using GTOs as basis functions, which are far from the maximally localized limit.

It is worth ending this section by briefly discussing the work of Lee et al. [9]. They present an expression for the VME which is, in fact, a particular case of our general expression Eq. (21) (we reproduce it in Appendix B). However, we believe that in order to reach their expression for the VME, they have inadvertently mixed Hilbert spaces. Their derivation starts from $\hat{\boldsymbol{v}} = i[\hat{H}, \hat{\boldsymbol{r}}]$ and, briefly, they follow by projecting $\langle n\boldsymbol{k}|\hat{\boldsymbol{v}}|n'\boldsymbol{k}\rangle_v = i\langle n\boldsymbol{k}|[\hat{H}, \hat{\boldsymbol{r}}]|n'\boldsymbol{k}\rangle_v$, expanding eigenstates in a non-orthonormal local orbital basis, and inserting the closure relation $\hat{I} = \sum_{\alpha\boldsymbol{R},\alpha'\boldsymbol{R}'} S_{\alpha\boldsymbol{R},\alpha'\boldsymbol{R}'}$ between the product of operators. We note that their procedure is equivalent to start by writing

$$\langle n\boldsymbol{k}|\hat{\boldsymbol{v}}|n'\boldsymbol{k}\rangle_v = i\omega_{n\boldsymbol{k},n'\boldsymbol{k}}\langle n\boldsymbol{k}|\hat{\boldsymbol{r}}|n'\boldsymbol{k}\rangle, \tag{28}$$

and proceeding in the same manner. The problem of starting with Eq. (28) is that, as explained in Sec. 2, this equality only holds in the case of open boundary conditions (infinite systems). This is not the case when using Bloch states constructed as a phased sum of local orbital basis sets, namely, Wannier orbitals or TB models, where the band eigenstates obey a finite volume normalization [see Eq. (5) and Eq. (22)]. The correct result found in Ref. [9] can only be explained by the unjustified identification of $\langle n\boldsymbol{k}|\hat{\boldsymbol{r}}|n'\boldsymbol{k}\rangle_v + \boldsymbol{C}_{n\boldsymbol{k},n'\boldsymbol{k}'}$ in Eq. (11) with $\langle n\boldsymbol{k}|\hat{\boldsymbol{r}}|n'\boldsymbol{k}\rangle = \int_{-\infty}^{\infty} d^3r\, \psi_{n\boldsymbol{k}}^{(v)*}(\boldsymbol{r})\boldsymbol{r}\psi_{n'\boldsymbol{k}}^{(v)}(\boldsymbol{r})$, which leads to Eq. (28). By doing this, the dependence on an arbitrary origin of integration is effectively removed by the new integration limits, but the integral is ill-defined. This subtle issue, which can be easily missed, is stressed by our notation in Eq. (28), where we have put the subscript $v$ is on the left hand side but not on the right hand side of the equality.

In the next section we present some numerical examples in order to explore the details of the VME formulas in a practical situation.

## 4 Practical cases: hexagonal boron nitride and graphene

The first goal of this section is to gauge the importance of the different terms in Eq. (7), by comparing between independent evaluations of the VME and MME. Particularizing to a local orbital basis case, the former can be evaluated from Eq. (21), while the latter becomes $\langle n\boldsymbol{k}|\hat{\boldsymbol{p}}|n'\boldsymbol{k}\rangle = -i\sum_{\alpha\alpha'\boldsymbol{R}} c_{\alpha n}^*(\boldsymbol{k})c_{\alpha'n'}(\boldsymbol{k})e^{i\boldsymbol{k}\cdot\boldsymbol{R}}\langle\alpha\boldsymbol{0}|\nabla_{\hat{\boldsymbol{r}}}|\alpha'\boldsymbol{R}\rangle$. Our second goal is to explore the relative importance of the two terms in Eq. (21).

### 4.1 Detailed numerical analysis of VME and MME

We start by computing the band structure of a benchmark material. We choose a monolayer of hexagonal boron nitride (hBN), which is a sufficiently complex system to our purposes. In Fig. 1, we show: (i) a tight-binding (TB) two-band calculation for the upper (lower) valence (conduction) bands, including only first neighbour interactions between the $p_z$ orbitals of B and N atoms, (ii) a DFT calculation employing a small-core pseudopotential basis set [14] to replace the $1s^2$ electrons in every atom (labelled here as CRENBL [14])[2] and, (iii) an all-electron calculation with the 6-31G* basis set [15]. The DFT calculations were performed using CRYSTAL17 [16] with the local von Barth-Hedin exchange-correlation functional [17].

---

[2]Our basis sets can be found on https://www.basissetexchange.org/ by searching in the element's database (as accesed on October 2022).

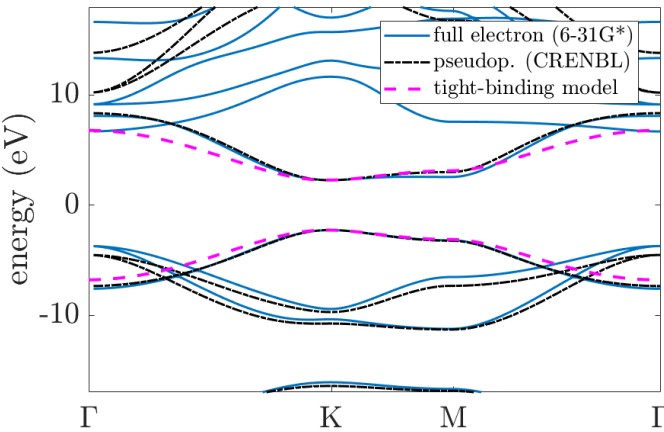

Figure 1: Comparison of the band structure of monolayer hBN using different approaches: (i) a first-neighbour tight-binding two-band model with 2.15 eV hopping, (ii) a DFT calculation using a small-core pseudopotential basis set and (iii) a DFT all-electron calculation (see text for further details).

For our further analysis, we require to perform matrix elements in the form $\langle \gamma^{(\alpha_i)}_{lm} | \hat{O} | \gamma^{(\alpha_i')}_{l'm'} \rangle$, with $\gamma^{(\alpha_i)}_{lm}(\boldsymbol{r})$ GTOs basis sets [16] centered at the atomic sites ($l$ and $m$ give the symmetry of the orbital while $\alpha_i$ its spatial extent). For momentum and position operators, we are left with integrations that can be evaluated in an analytical fashion.

We are not concerned here with the accuracy of the obtained gap so we have excluded the use of hybrid functionals and their possible extra non-local contributions.

Both DFT band structures are essentially similar up to the conduction band. The agreement is particularly good for the valence and conduction bands, both with a band gap of 4.55 eV, except maybe for a noticeable difference at the M point of $\simeq 0.5$ eV. As expected, only the more accurate all-electron calculation with a large basis can reproduce results in the literature [18]. The tight-binding parameters can be fitted to resemble one of these calculations. It is easier to obtain a better overall fit to the CRENBL band structure with a hopping $t = 2.15$ eV, as only two $p_z$ orbitals are present to reproduce the energy dispersion.

We now explore in some detail Eqs. (7) and (21). To this purpose, in Fig. 2 we show the magnitude of several quantities relevant to the band-gap optical transition along the $\Gamma - K - M$ path. Fig. 2(a) shows the absolute value of the $x$ component of the VME and the MME for the three cases shown in Fig. 1. Looking at the VME, the TB result deviates quantitatively from the other two, but not qualitatively. When comparing the VME and the MME, we observe that for the CRENBL basis the difference is significant, particularly near M, while that for the large basis this difference is negligible. We explain this differences as follows. For the CRENBL case, this difference comes, as reflected in Eq. (7), due presence of the non-local pseudopotential, invalidating the velocity and momentum equivalence. In the all-electron case, the difference is almost negligible. Since the evaluation of the MME is essentially analytical due to the use of Gaussian orbitals, we discard possible errors when evaluating Eq. (21). Hence, we attribute the very small difference to the finite size of the Hilbert Space, which is always required in numerical calculations [see our discussion below Eq. (7)]. This effect is, of course, also present in the pseudopotential case and should be more important due to the smaller Hilbert Space (8 and 36 bands for the CRENBL and all-electron case, respectively.)

We have therefore notice the use of non-local functionals in the starting DFT calculation. In order to explore this effect further in, we have repeated the all electron (6-31G*) calcula-

tion using the HSE06 hybrid functional [19]. We have also used the same large all-electron basis as before, allowing us to isolate the effect of the non-locality from that coming from the finite basis size. We show the results in the inset of Fig. 2(a), where we compare the velocity and momentum curves. In this case, the deviation between the VME and MME curves becomes appreciable, but not larger than the one stemming from the finite basis in the previously discussed CRENBL case. Our results suggest that the use of a nonlocal functional have a relevant impact when assuming the equality versus momentum and velocity operators. In summary, these results explicitly show that the VME and MME cannot always be taken as the same quantity. This can only be safely done, in principle, when using large all-electron basis sets in DFT-LDA calculations.

In Fig. 2(b), we compare the magnitude of the $k$-gradient term [the A term in Eq. (21)] calculated in the atom gauge for the three different cases. In the TB case, this term gives the full value for the VME. In the DFT case, the results deviate significantly from the exact value [shown in Fig. 2(a)], showing the importance of the B term in Eq. (21). The CRENBL basis presents a larger deviation: this shows that having a smaller size do not guarantees having small inter-orbital contributions to Eq. (25), which turn into an important contribution to $v^{(B,\text{atom})}$. As mentioned in Sec. 3, only the maximal localization condition (or point-like orbitals) for the basis set ensures that $v^{(A,\text{atom})}$ gives the exact VME. This condition is not met in neither of the two DFT basis sets used in our calculations. We also show the result obtained in the cell gauge in Fig. 2(c). Now, not only quantitative differences appear, but also selection rules break when approaching the $\Gamma$ point (here the VME must be zero according to the irreducible representations of the wave functions). Therefore, identifying the VME simply as a $k$-gradient of the Bloch Hamiltonian in the cell gauge can lead, not only to quantitative errors, but also to incorrect physical interpretations.

## 4.2 Optical conductivity

The calculation of an experimentally measurable quantity such as the optical conductivity can be affected by an incorrect evaluation of the VME. To show this we make use of the Kubo-Greenwood [20] expression (we do not use atomic units here for clarity):

$$\sigma_{\alpha\beta}(\omega) = -\frac{i}{N_k}\frac{e^2\hbar}{\Omega}\sum_{nn'k}\left(\frac{f_{nk}-f_{n'k}}{\omega_{nk,n'k}}\right)\times\frac{\langle nk|\hat{v}_\alpha|n'k\rangle_v\,\langle n'k|\hat{v}_\beta|nk\rangle_v}{\hbar\omega+\omega_{nk,n'k}+i\hbar\eta}, \qquad (29)$$

where $f_{nk}$ is the Fermi-distribution occupation number and $N_k$ is the number of $k$ points in the discretized Brillouin zone.

In Fig. 3 we show the longitudinal optical conductivity, computing the VMEs within the different approximations considered in previous section. We have separated the results obtained with the small-core basis from those with the all-electron basis, as shown in Fig. 3(a) and Fig. 3(b), respectively, where we have also added the calculation with the TB model in both panels. At the bandgap frequency, the DFT and TB calculations involving the exact VME are able to reproduce the quasiuniversal behaviour [21] for a parabolic noninteracting semiconductor, yielding $\sigma = e^2/2\hbar$. The use of MMEs, instead of the VMES, fails for the pseudopotential and small basis case [black dashed line in Fig. 3(b)], as expected from the discussion in previous subsection. At higher frequencies the TB model underestimates the response, which is similar in magnitude for both DFT cases, the only difference being the position of the Van Hove singularity which originates in the bands at the M point (see Fig. 1). In both DFT calculations, replacing the VME by its $k$-gradient approximation overestimates the exact result. A calculation with the $k$-gradient term in the cell gauge $\hat{v} \rightarrow v^{(A,\text{cell})}$ (not shown) gives an even larger discrepancy at all frequencies, but worse, also removes the isotropic behaviour of the conductivity tensor with $\sigma_{xx} \neq \sigma_{yy}$. This erroneous behaviour has been already discussed for

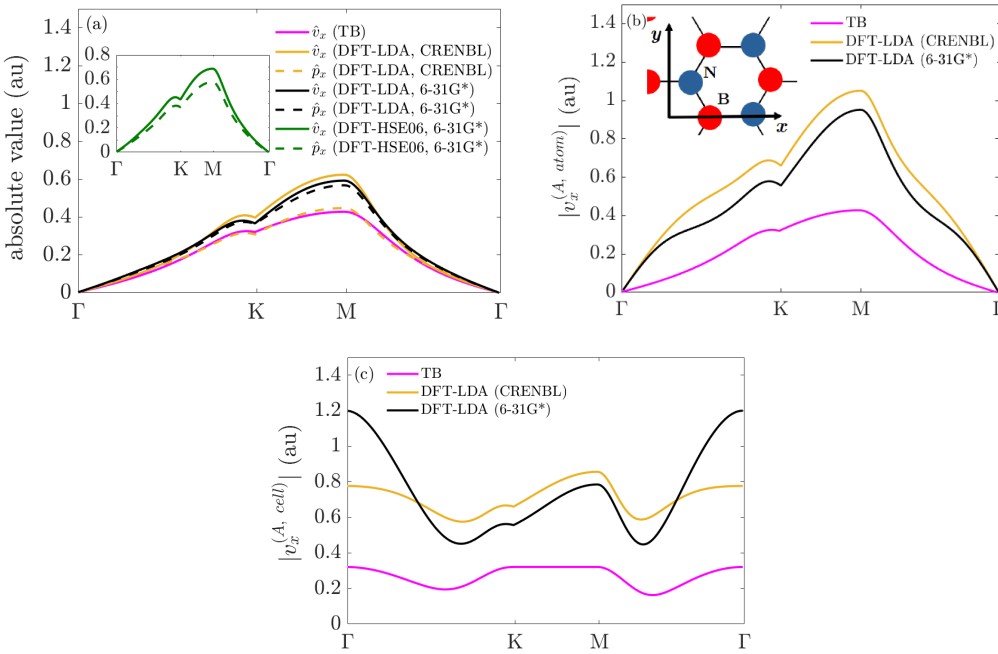

Figure 2: Absolute value of matrix elements for the band-gap transition along the Brillouin zone of monolayer hBN. (a) Velocity and momentum matrix elements for the two DFT calculations presented in 1 [the inset show an extra DFT case using the HSE06 functional (see main text)] and in the tight-binding approximation (b) Same for the first term of Eq. (21) in the atom gauge [see. Eq. (23)]. (c) Same as (b) but in the cell gauge.

graphene in Ref. [22] and highlights the importance of taking the *k*-gradient approximation for VME using the appropriate gauge. It is also worth mentioning here the work by Wissgott *et al.* [23]. There, the Peierls approximation in the atom gauge is tested versus the complete VME also through a conductivity analysis of transition-metal oxides. Our conclusion about the gauge choice, not explored in their work, could give a better insight about the discrepancies that are found in Ref. [23].

A direct comparison with experiments can be made by analyzing the optical response of graphene. It is known that monolayer graphene shows a quasi-constant absorbance of $\sim 2.3\%$, corresponding to $\sigma = e^2/4\hbar$, over the energy region that goes from the far-infrared to the visible spectrum where excitonic effects are negligible ($< 2$ eV) [24–26]. Therefore, in this energy range, Kubo-Greenwood DFT-based calculations are expected to give a faithful optical response. In Fig. 4 we show the optical conductivity calculated with two different basis sets, equivalent to those used for hBN. We present results for the exact VMEs and their approximated values using the MMEs. Experimental results from Ref. [24] are also shown. We can see that both basis sets give results in very good agreement with the experimental ones when employing VME. For the case of MMEs, the CRENBL basis set gives $\sim 0.175 e^2/\hbar$, which translates in a 30 % error when comparing to the experimental curve. This result complements our previous study of hBN, showing the significant effect of non-local operators and finite basis sets when trying to replace the VMEs by the MMEs.

We end this section by commenting the recent work by Ibañez-Aspiroz and coworkers [10], which has been carried out in parallel to our study. They have explored the effect of progressively adding inter-atomic position matrix elements in Eq. (27), through a Wannier interpolation scheme. In Ref. [10] it is found that including position matrix elements beyond orbital

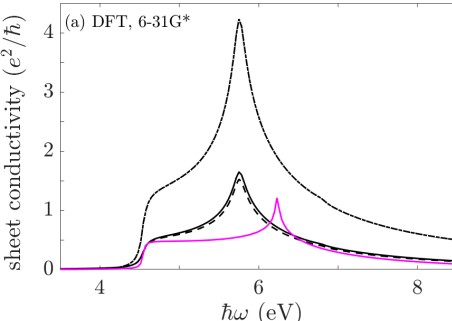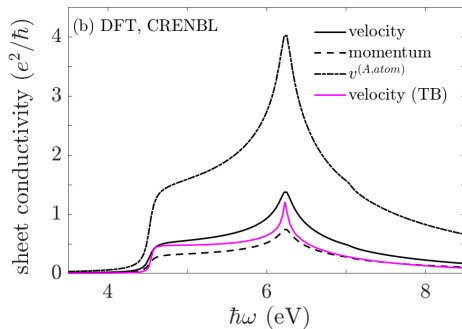

Figure 3: Frequency-dependent sheet conductivity of monolayer hBN as obtained from the evaluation of Eq. (29) using (a) a all-electron and (b) small-core pseudopotential DFT calculations. The calculation with a first-neighbour tight-binding model is included in both panels. See Fig. 1 for the corresponding band structures.

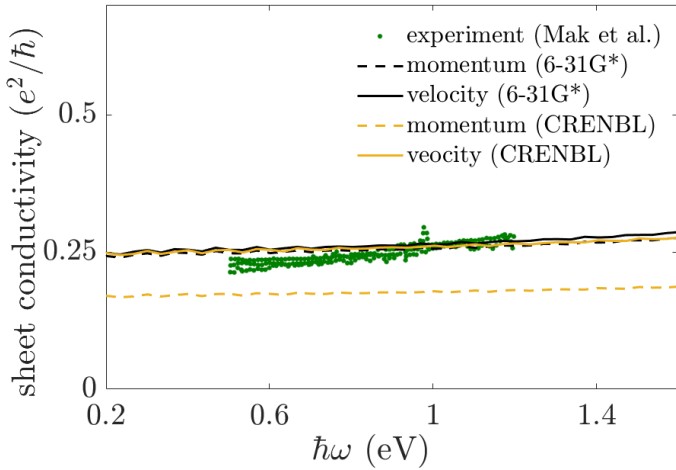

Figure 4: Same as Fig. 3 for the case of graphene. Experimentals results from Ref. [24] are shown. VME and MMe has been used to represent the velocity operator in the Kubo-Greenwood formula for the two DFT calculations (other approximations are not shown in this case).

centers in Eq. (25) leads to appreciable quantitative differences in the evaluation of the linear (dielectric function) optical response of $BC_2N$. Even more, it is shown that very significant errors are introduced when computing the quadratic response (shift photoconductivity). Here, we have shown that performing such approximation using GTOs as basis sets can lead to greater discrepancies even in the evaluation of the Kubo linear response. This is related to the fact that GTOs are less localized that Wannier functions, which are usually maximally localized per construction.

## 5 Conclusion

We have presented a comprehensive study of the evaluation of VME in crystalline solids, as obtained from the fundamental relation $\hat{v} = i[\hat{H}, \hat{r}]$. We have scrutinized several available expressions in the literature, filling the gaps and connecting them in a coherent story. We have seen that, when working in coordinate representation, one is bound to deal with a very inconvenient surface term which can be avoided by going first into the $k$-representation. We have

obtained a general expression which contains a familiar $k$-gradient term plus a correction term which involves the Berry connection of the Bloch basis elements. When using local orbitals as a basis, this can be rewritten in a more familiar form (see, e.g., Ref. [9]), but whose previous derivations contain unjustified mathematical steps. We have also shown several equivalences which involve the momentum and position operators, including well-known expressions in the crystal momentum representation (nonphysical distribution basis).

We have numerically tested the validity of different approximations to the VME by computing the optical conductivity of monolayer hBN and graphene through the Kubo-Greenwood formula. In particular, we have shown that approximating the VME by $\nabla_{\mathbf{k}} H_{\alpha\alpha'}(\mathbf{k})$ in a non-orthonormal basis produces significant quantitative errors and may also give rise to qualitative ones if one is not careful with the choice of gauge. We have also made emphasis on the fact that the velocity and momentum matrix elements can only be safely interchanged if the Hamiltonian is free of non-local terms and eigenstates are well-represented in the working Hilbert space.

Additionaly, our numerical analysis is in close relation to a very recent work of Ref. [10], where a similar study has been carried out using Wannier functions. We expect our work will contribute to remark the importance of going beyond the $k$-gradient approximation for the velocity matrix elements when using general local orbital basis sets. In summary, this work may well serve as a complete as well as a rigorous guide to the intricate relations behind the evaluation of the velocity, momentum, and position matrix elements in crystalline solids.

## Acknowledgments

The authors acknowledge financial support from Spanish MINECO through Grant No. PID2019-109539GB-C43, the María de Maeztu Program for Units of Excellence in R&D (Grant No. CEX2018-000805-M), the Comunidad Autónoma de Madrid through the Nanomag COST-CM Program (Grant No. S2018/NMT-4321), the Generalitat Valenciana through Programa Prometeo/2021/01, the Centro de Computación Científica of the Universidad Autónoma de Madrid and the computer resources of the Red Española de Supercomputación.

## A  Representation of $\hat{v} = i[\hat{H}, \hat{r}]$ in the distribution basis

In this appendix we prove that projecting $\hat{v} = i[\hat{H}, \hat{r}]$ in CMR along with the corresponding expression for the position operator, Eq. (15), allows to obtain Eq. (9). Let $|m\mathbf{k}\rangle$ be a general Bloch basis (orthonormal for simplicity) following a distribution normalization. An identification with the eigenstates basis will be made in the end. We have

$$\langle \phi_1 | \hat{v} | \phi_2 \rangle = i \langle \phi_1 | [\hat{H}, \hat{r}] | \phi_2 \rangle = i \sum_{mm'} \int_{\mathrm{BZ}} d^3k d^3k' g_m^{(1)*}(\mathbf{k}) g_{m'}^{(2)}(\mathbf{k}') \langle m\mathbf{k} | [\hat{H}\hat{r} - \hat{r}\hat{H}] | m'\mathbf{k}' \rangle \, .$$

(A.1)

Now we insert the closure relation between the two operators:

$$\begin{aligned}
\langle \phi_1 | \hat{v} | \phi_2 \rangle &= i \langle \phi_1 | [\hat{H}, \hat{r}] | \phi_2 \rangle \\
&= i \sum_{mm'm''} \int_{\mathrm{BZ}} d^3k d^3k' d^3k'' g_m^{(1)*}(\mathbf{k}) g_{m'}^{(2)}(\mathbf{k}') \\
&\qquad \times (H_{m\mathbf{k}, m''\mathbf{k}''} \mathbf{r}_{m''\mathbf{k}'', m'\mathbf{k}'} - \mathbf{r}_{m\mathbf{k}, m''\mathbf{k}''} H_{m''\mathbf{k}'', m'\mathbf{k}'}) \\
&\equiv \langle \phi_1 | \hat{v}^{(1)} | \phi_2 \rangle + \langle \phi_1 | \hat{v}^{(2)} | \phi_2 \rangle \, .
\end{aligned}$$

(A.2)

We have splitted the full matrix elements into two terms according to the two parts in Eq. (15). First we work out $\langle\phi_1|\hat{v}^{(1)}|\phi_2\rangle$,

$$
\begin{aligned}
\langle\phi_1|\hat{v}^{(1)}|\phi_2\rangle = &\frac{i}{\Omega}\sum_{mm'm''}\int_{\text{BZ}}d^3k\,d^3k'\,d^3k''\,g_m^{(1)*}(k)g_{m'}^{(2)} \\
&\times(k')H_{mm''}(k)\delta(k-k'')[-i\delta_{m''m'}\nabla_{k'}\delta(k'-k'')] \\
&-\frac{i}{\Omega}\sum_{mm'm''}\int_{\text{BZ}}d^3k\,d^3k'\,d^3k''\,g_m^{(1)*}(k)g_{m'}^{(2)} \\
&\times(k')[-i\delta_{mm''}\nabla_{k''}\delta(k''-k)]H_{m''m'}(k'')\delta(k''-k').
\end{aligned}
\tag{A.3}
$$

We have taken into account that a crystal Hamiltonian is diagonal in the $k$ vector, this is $H_{mk,m'k'}\equiv\Omega^{-1}H_{mm'}(k)\delta(k-k')$. Using the identity $F(k)[\nabla_k\delta(k-k')]=-[\nabla_k F(k)]\delta(k-k')$ straightforwardly, one can see

$$
\begin{aligned}
\langle\phi_1|\hat{v}^{(1)}|\phi_2\rangle = &\frac{1}{\Omega}\sum_{mm'}\int_{\text{BZ}}d^3k\,d^3k'\,d^3k''\,g_m^{(1)*}(k)g_{m'}^{(2)}(k')H_{mm'}(k)\delta(k-k'')[\nabla_{k'}\delta(k'-k'')] \\
&-\frac{1}{\Omega}\sum_{mm'}\int_{\text{BZ}}d^3k\,d^3k'\,d^3k''\,g_m^{(1)*}(k)g_{m'}^{(2)}(k')[\nabla_{k''}\delta(k''-k)]H_{mm'}(k'')\delta(k''-k') \\
= &-\frac{1}{\Omega}\sum_{mm'}\int_{\text{BZ}}d^3k\,g_m^{(1)*}(k)[\nabla_k g_{m'}^{(2)}(k)]H_{mm'}(k) \\
&+\frac{1}{\Omega}\sum_{mm'}\int_{\text{BZ}}d^3k\,g_m^{(1)*}(k)g_{m'}^{(2)}(k)[\nabla_k H_{mm'}(k)] \\
&-\frac{1}{\Omega}\sum_{mm'}\int_{\text{BZ}}d^3k\,d^3k'\,\nabla_k[g_m^{(1)*}(k)H_{mm'}(k)]g_{m'}^{(2)}(k')\delta(k-k'),
\end{aligned}
\tag{A.4}
$$

and applying the chain rule,

$$
\begin{aligned}
\langle\phi_1|\hat{v}^{(1)}|\phi_2\rangle = &-\frac{1}{\Omega}\sum_{mm'}\int_{\text{BZ}}d^3k\,\nabla_k[g_m^{(1)*}(k)g_{m'}^{(2)}(k)H_{mm'}(k)] \\
&+\frac{1}{\Omega}\sum_{mm'}\int_{\text{BZ}}d^3k\,g_m^{(1)*}(k)g_{m'}^{(2)}(k)[\nabla_k H_{mm'}(k)].
\end{aligned}
\tag{A.5}
$$

The first term is zero following the conditions required by Blount [11]. Now we look at the Berry connection term

$$
\begin{aligned}
\langle\phi_1|\hat{v}^{(2)}|\phi_2\rangle = &\frac{i}{\Omega^2}\sum_{mm'm''}\int_{\text{BZ}}d^3k\,d^3k'\,g_m^{(1)*}(k)g_{m'}^{(2)}(k')[H_{mm''}(k)A_{m''m'}(k'')\delta(k-k'')\delta(k'-k'') \\
&-A_{mm''}(k)H_{m''m'}(k'')\delta(k''-k)\delta(k''-k')] \\
= &\frac{i}{\Omega^2}\sum_{mm'm''}\int_{\text{BZ}}d^3k\,g_m^{(1)*}(k)g_{m'}^{(2)}(k')[H_{mm''}(k)A_{m''m'}(k)-A_{mm''}(k)H_{m''m'}(k)].
\end{aligned}
\tag{A.6}
$$

We now find the expression in the eigenstates basis. For clarity we rename $m=n$, and use $H_{nn'}(k)=\epsilon_n(k)\delta_{nn'}\Omega$, obtaining

$$
\begin{aligned}
\langle\phi_1|\hat{v}|\phi_2\rangle = &\sum_{nn'}\int_{\text{BZ}}d^3k\,g_n^{(1)*}(k)g_{n'}^{(2)}(k)\left\{\nabla_k\epsilon_n(k)\delta_{nn'}+\frac{i}{\Omega}[\epsilon_n(k)-\epsilon_{n'}(k)]A_{nn'}(k)\right\} \\
\equiv &\sum_{nn'}\int_{\text{BZ}}d^3k\,d^3k'\,g_n^{(1)*}(k)g_{n'}^{(2)}(k')\langle nk|\hat{v}|n'k'\rangle,
\end{aligned}
\tag{A.7}
$$

where $\langle nk|\hat{v}|n'k'\rangle = \delta(k-k')\big[\nabla_k\epsilon_n(k)\delta_{nn'} + i\Omega^{-1}\omega_{nk,n'k'}A_{nn'}(k)\big]$, which is precisely Eq. (16).

## B  Derivation of Eq. (21)

We start from Eq. (9) for the case $k = k'$,

$$\langle nk|\hat{v}|n'k\rangle_v = i\omega_{nk,n'k}A_{nn'}(k) + \nabla_k\epsilon_n(k)\delta_{nn'}. \tag{B.8}$$

Recall that the Berry connection is defined $A_{nn'}(k) \equiv i\langle u_{nk}|\nabla_k u_{n'k}\rangle_\Omega$. Expanding the periodic part in a Bloch Basis, $|nk\rangle$ state is $|u_{nk}\rangle = \sum_\alpha c_{\alpha n}(k)|u_{\alpha k}\rangle$, we readily obtain

$$A_{nn'}(k) = i\sum_{\alpha\alpha'} S_{\alpha\alpha'}(k)c^*_{\alpha n}(k)\nabla_k c_{\alpha'n'}(k) + \sum_{\alpha\alpha'} c^*_{\alpha n}(k)c_{\alpha'n'}A_{\alpha\alpha'}(k). \tag{B.9}$$

We now insert this expression into Eq. (B.8), obtaining

$$\begin{aligned}
\langle nk|\hat{v}|n'k\rangle_v = &-\epsilon_n(k)\sum_{\alpha\alpha'} c^*_{\alpha n}(k)\nabla_k c_{\alpha'n'}(k)S^{(v)}_{\alpha\alpha'}(k) \\
&+ \epsilon_{n'}(k)\sum_{\alpha\alpha'} c^*_{\alpha n}(k)\nabla_k c_{\alpha'n'}(k)S^{(v)}_{\alpha\alpha'}(k) + \nabla_k\epsilon_n(k)\delta_{nn'} \\
&+ i[\epsilon_n(k)-\epsilon_{n'}(k)]\sum_{\alpha\alpha'} c^*_{\alpha n}(k)c_{\alpha'n'}(k)A_{\alpha\alpha'}(k).
\end{aligned} \tag{B.10}$$

Applying the chain rule in the second term

$$\begin{aligned}
\epsilon_{n'}(k)\sum_{\alpha\alpha'} c^*_{\alpha n}(k)\nabla_k c_{\alpha'n'}(k)S^{(v)}_{\alpha\alpha'}(k) = &-\epsilon_{n'}(k)\sum_{\alpha\alpha'}\nabla_k c^*_{\alpha n}(k)c_{\alpha'n'}(k)S^{(v)}_{\alpha\alpha'}(k) \\
&-\epsilon_{n'}(k)\sum_{\alpha\alpha'} c^*_{\alpha n}(k)c^*_{\alpha n'}(k)\nabla_k S^{(v)}_{\alpha\alpha'}(k),
\end{aligned} \tag{B.11}$$

so we have

$$\begin{aligned}
\langle nk|\hat{v}|n'k\rangle_v = &-\sum_{\alpha\alpha'}\epsilon_n(k)S^{(v)}_{\alpha\alpha'}(k)c^*_{\alpha n}(k)\nabla_k c_{\alpha'n'}(k) - \sum_{\alpha\alpha'}\nabla_k c_{\alpha n}(k)\epsilon_{n'}(k)S^{(v)}_{\alpha\alpha'}(k)c^*_{\alpha'n'}(k) \\
&-\epsilon_{n'}(k)\sum_{\alpha\alpha'} c^*_{\alpha n}(k)c_{\alpha'n'}(k)\nabla_k S^{(v)}_{\alpha\alpha'}(k) + \nabla_k\epsilon_n(k)\delta_{nn'} \\
&+ i[\epsilon_n(k)-\epsilon_{n'}(k)]\sum_{\alpha\alpha'} c^*_{\alpha n}(k)c_{\alpha'n'}(k)A_{\alpha\alpha'}(k).
\end{aligned} \tag{B.12}$$

Now we can introduce the Hamiltonian matrix elements in the first two terms according to the eigenvalue equation, yielding

$$\begin{aligned}
&-\sum_{\alpha\alpha'} H^{(v)}_{\alpha\alpha'}(k)c^*_{\alpha n}(k)\nabla_k c_{\alpha'n'}(k) - \sum_{\alpha\alpha'}\nabla_k c^*_{\alpha n}(k)H^{(v)}_{\alpha\alpha'}(k)c_{\alpha'n'}(k) = \\
&-\nabla_k\bigg[\sum_{\alpha\alpha'} c^*_{\alpha n}(k)H^{(v)}_{\alpha\alpha'}(k)c_{\alpha'n'}(k)\bigg] + \sum_{\alpha\alpha'} c^*_{\alpha n}(k)c_{\alpha'n'}(k)\nabla_k H^{(v)}_{\alpha\alpha'}(k).
\end{aligned} \tag{B.13}$$

The first term cancels the gradient of the energy band in Eq. (B.12). In order to write the final form of the expression, we note that $A_{\alpha\alpha'}(k) \equiv i\langle u_{\alpha k}|\nabla_k u_{\alpha'k}\rangle_\Omega = i\nabla_k S^{(v)}_{\alpha\alpha'}(k) + A^*_{\alpha'\alpha}(k)$, which leave us with

$$\begin{aligned}
\langle nk|\hat{v}|n'k\rangle_v &= v^{(A)}_{nn'}(k) + v^{(B)}_{nn'}(k); \\
v^{(A)}_{nn'}(k) &= \sum_{\alpha\alpha'} c^*_{\alpha n}(k)c_{\alpha'n'}(k)\nabla_k H^{(v)}_{\alpha\alpha'}(k), \\
v^{(B)}_{nn'}(k) &= \sum_{\alpha\alpha'} c^*_{\alpha n}(k)c_{\alpha'n'}(k)\big[i\epsilon_n(k)A_{\alpha\alpha'}(k) - i\epsilon_{n'}(k)A^*_{\alpha'\alpha}(k)\big],
\end{aligned} \tag{B.14}$$

as presented in the main text. Finally this expression is particularized for Bloch states expanded in a local orbital basis, where $|\alpha k\rangle_\nu = 1/\sqrt{N}\sum_R e^{ik\cdot R}|\alpha R\rangle_\nu$, leading to the Berry connection

$$A_{\alpha\alpha'}(k) = \sum_R e^{ik\cdot R}\langle\alpha 0|\hat{r}|\alpha' R\rangle - \sum_R e^{ik\cdot R}R\langle\alpha 0|\alpha' R\rangle. \tag{B.15}$$

The expression above is a generalization for that of a Wannier basis, see e.g. Ref. [13,27]. Also note that $-\sum_R e^{ik\cdot R}R\langle\alpha 0|\alpha' R\rangle = i\nabla_k S_{\alpha\alpha'}^{(\nu)}(k)$. Here, an extra term arises accounting from the nonorthonormal character of atomic states, differently from the Wannier orbitals, which are orthonormal by construction. This is also reflected by the appearance of the overlap matrix in the first line of Eq. (20). In Eq. (B.15), dipole matrix elements between the basis set are integrated in all space and not in the unit cell, different than in the original definition for the Berry connection. This change is done by passing from $\int_{\text{cell}}$ to $\lim_{N\to\infty}\frac{1}{N}\int_{-\infty}^{\infty}$, that is well-defined for a periodic integrand. Finally Eq. (B.14) can be written

$$\begin{aligned}\langle nk|\hat{v}|n'k\rangle_\nu = &\sum_{\alpha\alpha'}c_{\alpha n}^*(k)c_{\alpha'n'}(k)\Big[\nabla_k H_{\alpha\alpha'}^{(\nu)}(k) - \epsilon_n(k)\nabla_k S_{\alpha\alpha'}^{(\nu)}(k)\Big] \\ &+ i[\epsilon_n(k) - \epsilon_{n'}(k)]\sum_{\alpha\alpha'}c_{\alpha n}^*(k)c_{\alpha n'}^*(k)\sum_R e^{ik\cdot R}\langle\alpha 0|\hat{r}|\alpha' R\rangle,\end{aligned} \tag{B.16}$$

which is the formula given in Ref. [9].

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
