# Peer review of "A comprehensive study of the velocity, momentum and position matrix elements for Bloch states: application to a local orbital basis"

_SciPost Physics Core, doi:SciPost Phys. Core 6, 002 (2023)_

## Round 1 · Referee Report · Anonymous (Referee 1) · 2022-9-15

Strengths

The authors set out to present a comprehensive theory of the velocity operator in the electronic structure theory of crystalline materials. 1- It is true that there is a long history of confusion on this topic, especially as it applies to incomplete basis sets, so I think this goal is worthy. 2- Moreover, I think there are some interesting general results in the paper. 3- For example, I think the general framework of Eq.~(21) is valuable. 4- Also, the numerical tests in Sec.~IV are of practical interest.

Weaknesses

For a paper that tries to dispel some of this previous confusion, the bar is raised for the level of clarity and pedagogical value in the present paper. Here I think it is still in need of improvement. I enumerate my concerns in the PDF attachment

Report

I think this manuscript may be suitable for publication in SciPost if the suggested clarifications can be made.

Requested changes

See enumerated comments in the PDF attachment.

Attachment

  • validity: high
  • significance: good
  • originality: good
  • clarity: high
  • formatting: excellent
  • grammar: excellent

Author:  Juan José Esteve-Paredes  on 2022-10-08  [id 2903]

(in reply to Report 1 on 2022-09-15)
Category:
answer to question

We are very grateful to the referee for the thorough and extensive review work. Here we address all his/her comments in the following. We hope that the concomitant modifications we have introduced in our manuscript have turned it into a more clear and rigorous work, and also more accesible to read.

We also make a resubmission including changes motivated by our referee's suggestions, as well as other minor changes.

1. I don't really understand why the manuscript puts so much emphasis on the result of Gu et al., Eq. (1). I was unfamiliar with this result, and had a hard time understanding at first what it was about.

The work by Gu et al. deals with the commutator formula for velocity (or momentum, as they consider) following a finite volume normalization (PBCs) and a representation using coordinate space. The main conclusion there is that momentum and dipole matrix elements (multiplied by the frequency) should never be interchanged when dealing with Bloch eigenstates in a finite volume. We consider that such reference treats excelently an important part needed to understand the formalism of momentum and position operators among the vast literature in this topic. Therefore, we felt it had to be sufficiently covered in our work.

**1.(a) The sentence below Eq. (1) suggested to my mind a finite crystallite ofvolume $v$ with physical surfaces, so I had trouble understanding why the eigenstates were not ``decaying at the boundaries" (vacuum tails). The authors should explain at the outset that the volume in question is just a mathematical construction of an $N \times N \times N$ supercell inscribed in the bulk. But just below Eq. (1) and later in Sec. II.D, it is clear that this expression is not true for general wave functions, or even for arbitrary functions of Bloch form; it is only true for Bloch states (eigenstates of $\hat{H}$). Also, it seems only to be true for $\hat{H}' = 0$ (absence of nonlocal terms in the potential). The authors also point out that it involves an awkward set of surface integrals. I wonder if this entire discussion could be deemphasized, or moved to an Appendix. **

As pointed out by the referee, Eq. (1) is derived assuming periodic boundary conditions for eigenstates, which implies an integration domain of volume $v$ when dealing with scalar products. We have rewritten the comment below Eq. (1) in order to clarify this detail. Regarding the second point, we feel that is appropiate to mantain the discussion in Sec. II.D in the main text as we think it greatly contributes to the goal of Sec. II. This is, explaining in a comprehensive manner the different approaches to deal with the relations between the velocity, momentum and position operators.

1.(b) I have a few comments about Eqs. (11-12) and the surrounding discussion. First, isn't it true that $\langle n\boldsymbol{k}|\hat{\boldsymbol{v}}|n'\boldsymbol{k}'\rangle_v$ in Eq. (1) contains a $\langle n\boldsymbol{k}|\hat{\boldsymbol{v}}|n'\boldsymbol{k}\rangle_v$? Why don't I see this reflected in Eqs. (10-12) Shouldn't we be writing expressions for quantities like $\langle n\boldsymbol{k}|\hat{\boldsymbol{v}}|n'\boldsymbol{k}\rangle_v$ instead? Are the two terms in Eq. (11) individually nonzero in Eq. (11), but they cancel for $\boldsymbol{k} \neq \boldsymbol{k}'$? I find it confusing.

Eqs. (11) and (12) are derived by directly projecting in the coordinate space of volume $v$. With this procedure, it is not possible to factorize out a kronecker delta and directly see the diagonal behaviour. As said in the question, both terms can be individually nonzero but they cancel for transitions between states with different crystal vectors. This is explicitely shown in Ref. [4] (Gu et al.) with an example. On the other hand, this diagonal behaviour can be anticipated when using the cell-periodic part of the eigenstates, see Eq. (8). In order to remark and clarify this issue, we have added a comment above Eq. (6) [recall that Eq. (8) is just an application of this] explaining how the wave-vector delta appears, as well as slightly enlarged the discussion below Eq. (12).

**1.(c) Also, can any intuitive physical interpretation be given to Eqs. (11) and (12)? The latter looks like a current flow across the boundary, doesn't it? I suppose that for diagonal elements $\langle n\boldsymbol{k}|\hat{\boldsymbol{v}}|n\boldsymbol{k}\rangle_v$, the first term in (11) vanishes, right? **

At a first sight, one surely thinks that the first term in Eq. (11) gives the contribution to velocity in the bulk while the second is the flow though the boundaries. However, recall that this is just a consequence of considering a finite volume and wavefunction. Therefore, we consider that such statement can be naive idea which may be far from a clear physical interpretation (contrarily as it happens in fluid mechanics, for instance), which is hard to do at the level of microscopic matrix elements. Gu et al. (where the equation is presented) do not mention anything about this detail, probably because no conclusions can be linked to this form of the equation. We have therefore avoided to make any comment that can potentially be a bit misleading. Regarding the second point, yes, as $\omega_{n\boldsymbol{k},n\boldsymbol{k}}=0$, all the contribution comes from the surface term. In this case one has that $\boldsymbol{C}_{nk,nk}=\partial_k \epsilon_n (\boldsymbol{k})$, accordingly to Eq. (9). We have included a discussion of this diagonal term in Sec. II.D.

1(d) I suspect Eqs. (11-12) can be recast in terms of integrals over a the interior and the boundary of a single primitive cell, with wave functions normalized to the unit cell, and with an average over $k$, where $k$ runs (as now) over the $N \times N \times N$ k-space mesh of the primitive Brillouin zone. It seems to me that this would be a more natural way to present the result.

One can try to reduce the integrals in the unit cell and see what happens. For instance, in a one dimensional model, the vertex of the integration line of length $L$ can be written $x_0=\eta a+\Delta$, where $\eta$ is an integer and $\Delta \in (-a/2,a/2)$. This serves to connect $x_0$ to a point inside the unit cell. We directly obtain \begin{equation} \begin{split} &\int_{x_0}^{x_0+L}dx\psi_{nk}^{(v)\ast}(x)x\psi_{n'k'}^{(v)}(x)= \ & \ \ \ \ \ e^{i(k'-k)\eta a}\int_{\Delta}^{\Delta+L}dx\psi_{nk}^{(v)\ast}(x)x\psi_{n'k'}^{(v)}(x) \ & \ \ \ \ \ +\eta a e^{i(k'-k)\eta a}\delta_{nn'}\delta_{kk'} \end{split} \end{equation} And similarly for the surface term. However, not much else can be done: as the integrand contains a non-periodic operator, we cannot reduce the integration over the whole space by a sum of integrals within a unit cell.

1.(e) Minor additional comment about Eq. (12): the final $\boldsymbol{r}$ is difficult to notice, but is crucial. Maybe there could be a way to call the reader'sattention to its presence.

We agree in this minor points and have added a sentence below Eq. (12) to remark it.

2. I found Eq. (7) and its discussion to be confusing. It seems to me that Eq. (7) is simply true without the last correction term. I think the authors have to find a way to be more explicit about what they really mean. For example, let $P_b$ be the projection operator onto the basis (``almost unity"), and define objects like $\hat{H}'=P_b \hat{H} P_b$, $\hat{p}'=P_b \hat{p} P_b$, $\hat{r}'=P_b \hat{r} P_b$, $\hat{v}'=P_b \hat{v} P_b$, etc. In this notation, I think what the authors are trying to say is that if the operators are replaced by their primed versions in Eq.(7), the $\Delta$ correction term is needed. Is that the idea? It needs to be clearer. By the way, I think there is a typo 7 lines below Eq. (7), where it should be Tr($r_\alpha, p_\beta)$=Tr($p_\beta,r_\alpha$) (when using primed operators in my notation).

We agree with the appreciation that such discussion needs to be clearer. As this effect has visible consequences in Figs. (2) and (3), we have kept the discussion but removed the delta term in the equation. The effect of the finite basis is discussed without any extra mathematical definitions for simplicity in the reading. We have also corrected the typo.

3. There is some confusing notation in Eq. (5). Putting $O(\boldsymbol{r})$ inside the integrand seems to suggest the $O$ is a local operator, but clearly this is not the case, or else $\hat{O}_{\boldsymbol{k}}$ would be the same as $\hat{O}$. I suppose what is meant would correspond to replacing

$$ \hat{O}(\boldsymbol{r})\psi_{n'k'} $$
by ** or more pedantically, by**
$$ \langle \boldsymbol{r}|\hat{O}|\psi_{n'k'}\rangle. $$

We understand the confusion and have rewritten Eq. (5) with the suggested notation.

4. Shortly after Eq. (5), the wording ¨such that still satisfies the Bloch theorem" is misleading. There was at least one more similar misuse later in the manuscript. Bloch's theorem is a theorem about the eigenstates of a Hamiltonian, which this wave function is not. The correct wording would instead be something like ¨of Bloch form". Even better would be to discuss whether $\hat{O}$ is a periodic operator, i.e., one that commutes with crystal translations; if so, then the product of such an operator with a wave function of Bloch form is automatically a wave function of Bloch form.

We also agree with this point and have used the term ``of Bloch form", which seems a natural way for us to describe those waves functions.

5. Another awkward detail of notation is the appearance of superscripts $(v)$ and $(n)$ in Eq. (19) and elsewhere. It looks as though the notation is parallel, whereas in fact the meanings of these superscripts are completely different.

We have made the change $c_{\alpha}^{(n)} (\boldsymbol{k}) \rightarrow c_{\alpha n}(\boldsymbol{k})$ to avoid posible confusions.

6. I think the LHS of the first Eq. (25) should be $\boldsymbol{A}_{\alpha \alpha'} (\boldsymbol{k})$. Also, the asymmetry between the two Eqs. (25) ¨looks wrong" although I think it is actually correct. It may be worth a few words of explanation. Perhaps it may be worth emphasizing that in the case of a nonorthogonal basis, A is not Hermitian, as it is in the orthonormal case. It also may be a good idea to explicitly write expression for the last term in Eq. (44) in a language parallel to the first term; I guess it comes to something like **

$$ \sum_{\boldsymbol{R}}e^{i \boldsymbol{k} \cdot \boldsymbol{R}} \boldsymbol{R} \langle \alpha\boldsymbol{0}|\alpha' \boldsymbol{R}\rangle, $$
** doesn't it?

We have added some commentary below Eq. (25) to highlight this points. Also, we agree and have done the second suggested change and identified this quantity with the gradient of the overlap matrix below Eq. (44).

7. The choice of the acronym CREN is not explained; what does it stand for?

To be honest: we do not really know. These basis set, originally introduced by Pacios and Christiansen (Ref. [15]), have stablished as one of the standard Gaussian-type basis sets used in Quantum Chemistry. They are usually labelled as CREN-BL or CREN-BS for large or small core pseudopotentials. For instance, the known database https://www.basissetexchange.org/ for GTO basis sets include the ``CREN-BL(BS)" basis for most elements. The information is found by clicking an element and looking in the list of available basis sets. We have included a reference to this online database page for completeness in our manuscript.

---

## Round 2 · Referee Report · Anonymous (Referee 1) · 2022-10-10

Strengths

See previous report. Overall I judge that the authors have done a conscientious job of replying to my comments and making corresponding changes to the manuscript. I have additional comments concerning my comments 1(b) and 1(d) that are detailed in the attachment, which may lead to optional revisions.

Weaknesses

See previous report

Report

I think the manuscript now meets the journal's standards and should be accepted for publication.

Requested changes

See attached PDF.

Attachment

  • validity: high
  • significance: good
  • originality: good
  • clarity: high
  • formatting: excellent
  • grammar: good

Author:  Juan José Esteve-Paredes  on 2022-10-14  [id 2921]

(in reply to Report 1 on 2022-10-10)

We acknowledge the referee the recommendation for publication. We also include below a small response to the last comment by our referee. We count the equations the referee's last response from 1 to 4, as seen in the webpage for the manuscript submission.

Overall I judge that the authors have done a conscientious job in replying to my comments and making corresponding changes in the manuscript. I still have my reservations about the way my comments 1(b) and 1(d) were answered. (Incidentally, the authors misquoted 1(b) with a cut-and-paste typo; the sentence should have read ¨First, isn't it true that $\langle \boldsymbol{k}|\boldsymbol{v}|n'\boldsymbol{k} \rangle_v$ in Eq. (1) contains a $\delta_{\boldsymbol{k}\boldsymbol{ k}'}$?".) If we accept from other arguments that $\langle n \boldsymbol{k}|\boldsymbol{v}|n'\boldsymbol{k}'\rangle_v$ is diagonal in $\boldsymbol{k}$ (e.g., using that $\boldsymbol{v}$ is a periodic operator), and in view of the $\omega_{n\boldsymbol{k},n'\boldsymbol{k}}$ prefactor, we only have to evaluate $\langle \boldsymbol{k}|\boldsymbol{r}|n'\boldsymbol{k} \rangle_v$ for $\boldsymbol{k}=\boldsymbol{k}'$ and $n\neq n'$. Here I adopt the notation that $\tilde{\psi}$ are the wave functions normaized to volume $L=Na$ while $\psi=\sqrt{N}\tilde{\psi}$ are normalized to a primitive cell. Then in 1D

$$ \begin{split} \langle \boldsymbol{k}|\boldsymbol{r}|n'\boldsymbol{k}\rangle_v&=\int_{0}^{Na}dx \ x \tilde{\psi}{nk}^{\ast}(x) \tilde{\psi}(x) \ &=\frac{1}{N}\int_{0}^{Na}dx \ x \psi_{nk}^{\ast}(x) \psi_{n'k}(x) \ &=\frac{1}{N}\sum_{j=0}^{N-1}\int_{ja}^{(j+1)a}dx \ x \psi_{nk}^{\ast}(x) \psi_{n'k}(x) \ &=\frac{1}{N}\sum_{j=0}^{N-1}\int_{0}^{a}dx (x+ja) \psi_{nk}^{\ast}(x) \psi_{n'k}(x) \end{split} $$

**where I have used that **

$$ \psi^{\ast}{nk}(x)\psi(x) $$

** is periodic under $x \rightarrow x+a$. Then**

$$ \begin{split} \langle n\boldsymbol{k}|\boldsymbol{r}|n'\boldsymbol{k} \rangle_v=& \Big( \frac{1}{N} \sum_{j=0}^{N-1}\Big) \int_{0}^{a}dx \ x \psi_{nk}^{\ast}(x) \psi_{n'k}(x) \ & \Big( \frac{a}{N} \sum_{j=0}^{N-1}j\Big) \int_{0}^{a}dx \psi_{nk}^{\ast}(x) \psi_{n'k}(x) \end{split} $$

The second integral vanishes by orthogonality of the wave functions and the first factor in the first term is unity, so

$$ \langle n\boldsymbol{k} |\boldsymbol{r}|n'\boldsymbol{k}\rangle_v=\int_{0}^{a}dx \ x \psi_{nk}^{\ast}(x) \psi_{n' \boldsymbol{k}}(x) $$

**I believe another argument along these lines allows to show that $C_{nk,n'k}$ can be similarly writen in terms of the boundaries of the primitive cell. This kind of development is that I had in mind when I wrote ¨I suspect Eqs. (11-12) can be recast in terms of integrals over a the interior and the boundary of a single primitive cell, with wave functions normalized to the unit cell''. I leave it as an option for the authors to discus this somehow in their revised manuscript. **

We thank the referee again for providing such detailed feedback. First, let us write the general version of Eq. (3) in this response for the general, nondiagonal case, (we use $\tau$ as the integration variable in the unit cell)

$$ \begin{split} \langle n\boldsymbol{k}|\boldsymbol{r}|n'\boldsymbol{k}' \rangle_v=& \delta_{kk'} \int_{\tau_0}^{\tau_0+a}d\tau \ \tau u_{nk}^{\ast}(\tau) u_{n'k}(\tau) \ & \Big( \frac{a}{N} \sum_{j=0}^{N-1}j \Big) \int_{\tau_0}^{\tau_0+a}d\tau \psi_{nk}^{\ast}(\tau) \psi_{n'k'}(\tau). \end{split} $$
Note that there is still a dependance with $\tau_0$, that can be related to the original $x_0$ origin for the integration volume $v$. Eq. (3) is a special case of this with $\tau_0=0$. Note also that the second term only includes the orthogonality condition for the $k$-diagonal case, in which we obtain the normalization condition for the periodic part: \begin{equation} \int_{\tau_0}^{\tau_0+a}d\tau \psi_{nk}^{\ast}(\tau) \psi_{n'k}(\tau)= \int_{\tau_0}^{\tau_0+a}d\tau u_{nk}^{\ast}(\tau) u_{n'k}(\tau) =\delta_{nn'}.
\end{equation} With this, we see that the simple form of Eq. (3) can only be obtained for the $k-$diagonal case, in which the position matrix elements still depend on the origin, as in the general case covered with Eqs (11) and (12) in the manuscript. The advantage of this reduction may be in terms of computational terms, where it could be desired to have a smaller integration grid. As nothing fundamental is added in this step, we prefer to avoid including this discussion in the manuscript.

---

## Round 2 · Author Response

Dear Editor,

Please find the resubmission of our manuscript after receiving the feedback of our referee. The changes in the text are written in red color.

Sincerely,

J.J. Esteve-Paredes and J.J. Palacios

---

## Round 2 · List of Changes

1. In the Introduction, we have enlarged the discussion below Eq. (1). (Referee's suggestion)
  2. In Sec. II.A, we have slightly changed the notation for the operators to avoid confusions regarding the local or nonlocal character of operators. We have also elaborated a bit more below Eq. (6). (Referee's suggestion)
  3. In Sec. II.B, we have removed the definition of a $\Delta$ term arising due to the incompletness of the Hilbert Space. Now we just comment this effect in the text, and also this paragraph has been moved to the end of the section. The discussion is maintained as it is relatable to our results in Section IV. (Referee's suggestion)
  4. In Sec. II.D, a minor comment has been added below Eq. (12), and the wording has been slightly changed. (Referee's suggestion)
  5. In Sec. II.D, the diagonal case for Eq (11) is now discussed. (Referee's suggestion)
  6. In Sec. III, a comment has been added below Eq. (25) to discuss Berry connection properties. (Referee's suggestion)
  7. In Sec. III, Ref. [15] was incorrectly cited below Eq. (27). It has been replaced by Ref. [8].
  8. In Sec. IV.A, we have changed the acronym CREN to CRENBL, as listed in the repository www.https://www.basissetexchange.org/. We have also include a citation to this URL.
  9. In Sec. IV.A, we have reacommodate the wording in the discussion based on the change number 3 in this list. Some other sentences has been rewritten in a clearer and relaxed way.
  10. In Sec. IV.A, the plots now include the previous notation change in the legends.
  11. In Sec. IV.A, an inset in Fig. 2(b) has be included showing how we arrange the atoms in our numerical calculations. The reader could now recreate our results easier.
  12. In Appendix B, the overlap matrix gradient has been explicitely shown in terms of the local orbitals. A identification with the former is made in the text below. (Referee's suggestion)

---

## Editorial Decision

published